# Advanced Low-Cost Technology for Assessing Metal Accumulation in the Body of a Metropolitan Resident Based on a Neural Network Model

**DOI:** 10.3390/s24227157

**Published:** 2024-11-07

**Authors:** Yulia Tunakova, Svetlana Novikova, Vsevolod Valiev, Maxim Danilaev, Rashat Faizullin

**Affiliations:** 1Department of General Chemistry and Ecology, Kazan National Research Technical University Named after A. N. Tupolev—KAI, Kazan 420111, Russia; yuatunakova@kai.ru; 2Department of Applied Mathematics and Computer Science, Kazan National Research Technical University Named after A. N. Tupolev—KAI, 10 K. Marx St., Kazan 420111, Russia; 3Research Institute for Problems of Ecology and Mineral Wealth Use of Tatarstan Academy of Sciences, 28 Daurskaya St., Kazan 420087, Russia; podrost@mail.ru; 4Department of Electronic and Quantum Information Transmission Systems, Kazan National Research Technical University Named after A. N. Tupolev—KAI, 10 K. Marx St., Kazan 420111, Russia; mpdanilaev@kai.ru; 5Institute of Fundamental Medicine and Biology, Kazan Federal University, Kazan 420008, Russia; rifajzullin@kpfu.ru

**Keywords:** metropolis, metals, entry into the body, retention of metals, health monitoring, long-term effects of metals on the body, short-term effects of metals on the body, multi-neural network model, open multi-neural network system, closed multi-neural network system, double-loop neural network system

## Abstract

This study is devoted to creating a neural network technology for assessing metal accumulation in the body of a metropolis resident with short-term and long-term intake from anthropogenic sources. Direct assessment of metal retention in the human body is virtually impossible due to the many internal mechanisms that ensure the kinetics of metals and the wide variety of organs, tissues, cellular structures, and secretions that ensure their functional redistribution, transport, and cumulation. We have developed an intelligent multi-neural network model capable of calculating the content of metals in the human body based on data on their environmental content. The model is two interconnected neural networks trained on actual measurement data. Since metals enter the body from the environment, the predictors of the model are metal content in drinking water and soil. In this case, water characterizes the short-term impact on the organism, and drinking water, combined with metal contents in soil, is a depository medium that accumulates metals from anthropogenic sources—the long-term impact. In addition, human physiological characteristics are taken into account in the calculations. Each period of exposure is taken into account by its neural network. Two variants of the model are proposed: open loop, where the calculation is performed by each neural network separately, and closed loop, where neural networks work together. The model built in this way was trained and tested on the data of real laboratory studies of 242 people living in different districts of Kazan. As a result, the accuracy of the neural network block for calculating long-term impact was 90% and higher, and the accuracy of the block for calculating short-term impact was 92% and higher. The closed double-loop model showed an accuracy of at least 96%. Conclusions: Our proposed method of assessing and quantifying metal accumulation in the body has high accuracy and reliability. It does not require expensive laboratory tests and allows quantifying the body’s metal accumulation content based on readily available information. The calculation results can be used as a tool for clinical diagnostics and operational and planned management to reduce the levels of polymetallic contamination in urban areas.

## 1. Introduction

A modern metropolitan resident is exposed to chronic exposure to excessive amounts of metals from various anthropogenic sources [1,2,3]. With excessive intake of metals, the body can mobilize internal reserves to a specific limit to maintain homeostasis. Still, after some time, an imbalance of metals in the body inevitably occurs. Metals exhibit toxicity if they accumulate in the body of a metropolitan resident. A sharp deterioration in the population’s health at the individual and population levels has been noted in places with an increased polymetallic background in environmental objects [4,5,6,7].

Metals entering the human body orally and with inhaled air accumulate in body biosubstrates. The most informative for assessment are those substrates involved in the transport, excretion, and accumulation of metals in the body [8,9,10]. The methodology of such studies is based on the theory of homeostatic balance [11,12]. The homeostasis mechanism of maintaining equilibrium concentrations of metals in body tissues has been substantiated by many studies [13,14,15]. The inertia of the mechanisms of homeostasis is manifested in a sharp increase in the proportion of ionized forms of metals at their excessive intake from the outside while increasing the excretory function of the kidneys and the concentration of elements in the urine [16,17,18]. The effect of “retention” of metals, which, in clinical practice, is accompanied by a decrease in their glomerular filtration rate, is called retention [19]. The complexity of measuring retention lies in the fact that its value is formed not only due to the various routes of intake and excretion of metals but also the characteristics of transport and deposition for the needs of the body, i.e., the balance of metals can be positive, negative, and neutral, and constantly changes its direction over a short period, which leads to contradictory results when using classical assessment methods [20,21,22]. At the same time, for adequate correction of microelement disorders in the body of a resident of a metropolis, a comprehensive assessment is necessary, characterizing the processes of intake, transport, and excretion of metals, taking into account the time interval [23].

Attempts to establish the priority pathways of various metals entering the human body from the environment have been made repeatedly. Still, such studies indicate that these pathways may be a priority depending on the conditions and exposure. On average, the food pathway of metal entry into the bodies of schoolchildren within a single territory can be accepted as a factor of constant and uniform action, and the quality of drinking water and air varies significantly, providing a significant gradient for the constantly acting background content of metals. Soil cover analysis characterizes the long-term formation of the level of air pollution with metals, covering a significant period. Moreover, the soil is a source of dust-containing metals, forming secondary air pollution in urbanized areas. Metals entering the human body orally and with inhaled air accumulate in their substrates. Therefore, one of the most reliable methods characterizing the polymetallic impact on the human body is assessing the metal content in substrates.

To date, assessing metal accumulation in the organism is based on expensive biological media and fluid analyses in laboratory conditions. The use of classical technologies associated with studying samples of biological body fluids is accompanied by many organizational, legal, material, and natural difficulties. Therefore, constructing models capable of calculating the content of metal accumulation in the body indirectly, based on easy-to-measure parameters not associated with invasive biological sampling procedures, is very relevant. We should note that retention is an integral indicator that summarises the multi-impact of external polluting factors and considers all pathways of metal intake into the organism [24,25,26,27]. We must take this fact into account in the calculation model.

Machine learning models, i.e., intelligent technologies, hold the most significant promise. The use of intelligent technologies to calculate retention in literature could be more frequent. Some works describe the calculation of metal content in environmental elements. Thus, in the work [28], the authors describe a neural network model that calculates metal contents in soil. Chinese researchers [29] consider a model of metal intake with the staple food, rice, and relate this to the metal content of soil based on a convolutional neural network. In [30], the biosorption of cobalt by Pseudomonas alkaliphilic NEWG-2 strain bacteria was based on a bilayer perceptron. Several other articles must cover the posed problem [31,32]. However, works are separate from the possibility of using intelligent neural network models to assess the level of metal retention in the population.

Our article aims to address this pressing issue by proposing a comprehensive model for calculating the metal accumulation (retention) level in the human body. This model considers not only the metal content in environmental elements (water, soil) but also the physiological characteristics of individuals. By considering both long-term and short-term metal intake from the outside, our model offers a holistic approach to the problem. Its potential to inform decisions on production reorganization, thereby preventing damage to public health, is a testament to its effectiveness.

## 2. Preliminary Information

The mathematical basis of the research in this article is artificial neural networks. An artificial neural network is one of the machine learning methods, a mathematical model that imitates the work of neurons in the human brain. The mathematical model is, as a rule, a distributed system capable of adaptation through analyzing factors affecting the neural network. In neural networks, individual computing modules, or nodes, are connected through weights adapted during the tuning (training) process to improve performance. The most straightforward node provides a linear combination of *n* weights $w_{1},\;\ldots ,\;w_{n}$, and *n* inputs $x_{1},\;\ldots ,\;x_{n}$, and passes the result through a nonlinear function φ, called the activation function (Figure 1).

The activation function determines the output values of neurons. The primary activation functions are as follows:
Sigmoid: $\varphi \left(x\right)=\frac{1}{1+e^{-x}}.$
Hyperbolic tangent: $\varphi \left(x\right)=\frac{e^{x}-e^{-x}}{e^{x}+e^{-x}}.$
Linear: $\varphi \left(x\right)=ax$ReLU: $R\left(x\right)=\max\left(0,x\right).$Threshold: $\varphi \left(x\right)=\left\{\begin{array}{c} 0,x\leq 0\\ 1,x>0 \end{array}\right.$SeLU: $S\left(x\right)=\left\{\begin{array}{c} \lambda x,\;x>0\\ \lambda (\alpha e^{x}-\alpha ),\;x\leq 0 \end{array}\right.$, λ≈1.0507; α≈1.6732

The graphs of the corresponding activation functions are shown in Figure 2.

Neural network models are defined by the network topology, node characteristics, and learning rules [33]. Regarding connection patterns, neural networks can be divided into two categories: feedforward networks, where graphs have no loops, and recurrent networks, where loops arise due to feedback (Figure 3).

Currently, neural networks are used to solve several problems, one of which is forecasting. Recurrent neural networks, such as Hopfield networks, are often used for forecasting time series [34]. ART models are used in classification problems, and Kohonen networks are used in clustering [35]. One of the most popular models for forecasts described as a regression problem is the multilayer perceptron (MLP), which consists of several layers of neurons and uses the backpropagation method for training [36]. The backpropagation method consists of adjusting the weights $w_{1},\;\ldots ,\;w_{n}$ of the neural network to minimize the loss function characterizing the network error. The basis of the method is the gradient descent procedure [37], which is widely used in optimization problems.

The idea of the procedure is as follows. Let a function f(x) be given, bounded below on the set $R^{n}$ and having continuous partial derivatives at all points of definition. It is required to find a local minimum of the function f(x) on the set of admissible solutions $X=R^{n}$, i.e., find a point $x^{*}\;\in R^{n}$ such that the following is true:$$f\left(x^{*}\right)=\underset{x\in R^{n}}{\min}f\left(x\right).$$

Let us construct a sequence of points $\left\{x^{k}\right\}$, k=0,1,2,… such that $f\left(x^{k+1}\right)<f\left(x^{k}\right)$, k=0,1,2,…. The points of the sequence are calculated according to the following rule:$$x^{k+1}=x^{k}-\alpha {f^{\prime }}^{\left(x^{k}\right)}$$
where α is the rate of gradient descent.

By replacing f(x) in the formula with the loss function and x with the weights $w_{1},\;\ldots ,\;w_{n}$ of the neural network layer, we can find such values of the weights that the loss function reaches a minimum. In this case, *k* denotes the training epochs. This study used a multilayer perceptron as a basic forecasting model. This choice is associated, on the one hand, with the relative simplicity of implementation and low consumption of computing resources compared to other types of networks and, on the other hand, with the nature of the problem being solved, which is a variant of the regression problem. In addition, the multilayer perceptron MLP is a universal approximator. This conclusion means that the appropriate setting of weights can approximate a function of any complexity. This fact allows us to use MLP to solve the problem where there are hidden non-formalizable dependencies.

## 3. Materials and Methods

We selected four metals for quantitative retention assessment: zinc, chromium, copper, and lead. These metals, known for their high variability in concentrations across the study area, were chosen due to their potential health implications in children and teenagers.

This study used the results of clinical tests of 242 children and teenagers living in different districts of Kazan obtained during the medical examination. Of these, 54 were boys (mean age 11.7 years) and 188 girls (mean age 10.4 years). To determine the metal content in biosubstrates, we carried out the following stages of sample preparation.

The blood was kept in a thermostat (37 °C) until a clot formed and then centrifuged for 15 min at 3000 rpm to separate the serum. The blood serum was diluted 1:2 with bidistilled water (to determine Zn, Cu, Fe). The blood serum was diluted with a filtrate obtained using trichloroacetic acid (TCA) to determine Pb, Cr, and Sr. Preliminary hydrolysis of serum proteins was carried out with hydrochloric acid (reagent grade). We added 0.75 mL of 1.5% HCl solution to 1.5 mL of blood serum. The solution was incubated for 1 h at 37 degrees. After hydrolysis of proteins, they were precipitated with 0.75 mL of 20% TCA, final dilution two times, and, after 1 h, centrifuged for 10 min at 1500 rpm. The supernatant liquid—TCA filtrate—was used for analysis. In cases where the metal concentration was below the detection limit, by the direct method or in the TCA filtrate, we used the extraction concentration method. For this purpose, we added 0.5 mL of 2% sodium diethyldithiocarbamate solution and two drops of TRITON X-100 detergent to 2.5 mL of serum; the mixture was vigorously shaken for 10 s. After 10 min, we added 1.5 mL of butyl acetate, and the mixture was shaken for 1 min and then centrifuged. Thus, it was possible to reduce the detection limit for chromium by 1.5 times and for strontium and lead by 2.5 times.

To determine metals in urine, a daily urine sample was thoroughly analyzed using a direct method. This comprehensive approach to analysis provides reliable data on metal concentrations in urine.

Water samples were taken at the outlet of the taps of houses’ internal water supply networks. In the sample preparation process, we evaporated 1 L of the analyzed drinking water in a water bath; the dry residue was dissolved in 50 mL of 1 N nitric acid (reagent grade), and the resulting aliquot was analyzed by atomic absorption spectrometry.

Extraction of gross forms of metals in soil samples was carried out with 5 N HNO_3_. A determination of the total amount of metal determinations was carried out: in drinking water, 1160, and, in soil cover, 1890.

The analytical measurement of metal concentrations was carried out using the advanced AAS method on an AAnalyst 400 atomic absorption spectrophotometer (PerkinElmer Inc., Waltham, MA, USA). The primary data were processed using the sophisticated AA WinLab32 software package version 7.3, demonstrating the technological sophistication of our research.

We constructed a comprehensive data matrix from the collected data. This matrix included information on metal content in various sources such as blood, urine, consumed drinking water, snow leachate, and soil cover in the area of residence. It also incorporated demographic factors like age, height, weight, body surface area, volume of daily diuresis, and sex.

Although the direct determination of metal retention in the human body is practically impossible, comparing the dynamics of the amount of incoming and excreted metal allows us to present a measure of its relative values. The body’s primary role in metal transport is assigned to the blood and the binding proteins. Metals in the blood in a molecularly dispersed state, in the form of ions or unstable complexes, are excreted mainly with urine. According to the ratio of excreted substances, the clearance of metals is estimated (CL*_M_*, mL/min) using the Reberg formula [38] (Equation (1)).
(1)$$CL_{M}=\frac{c_{M_{urine}}\times DM}{c_{M_{blood}}\times 1440}\times BS,$$
where $c_{M_{urine}}$, $c_{M_{blood}}$—metal concentration in urine and blood, respectively, µg/mL; *DM*—daily diuresis, mL; 1440—number of minutes daily; *BS*—body surface area corrected relative to the population average. The *BS* parameter is used as a correction for body surface area since the minute filtration volume in the kidneys depends on the height and weight of a person and requires normalization in people significantly deviating in size from the average values. The BS value is calculated according to Equation (2):(2)$$BS=\frac{1.73}{\sqrt{\frac{\mathrm{Weight}\times \mathrm{Height}}{3600}}},$$

Clearance reflects the rate of metal clearance from blood plasma: the lower the rate, the higher the metal retention. For the convenience of presentation and normalization of calculated values, retention is estimated using the exponential function of the ratio of absolute clearance values to their standard deviation (Equation (3)).
(3)$$Retentions_{M}={e^{-}}^{\frac{C{L_{M}}^{2}}{\sigma^{2}}},$$
where σ is the standard deviation of the data series.

All data were linked to the identification codes of the study participants and their places of residence. The collected measurement results were subjected to preliminary processing, including the following:Analysis of frequency characteristics of variation series with an assessment of the normality of data distribution;Data cleaning is based on statistical analysis and filtering out gross errors [39];Selection of the main influencing factors based on the methods of factor and component analysis [40];Data normalization using the Z-score normalization method based on the mathematical expectation and variance of the series, which, unlike other methods, allows for avoiding abnormal clustering of values near zero [41].

Preprocessing was performed using the TIBCO Data Science/Statistica statistical data processing package (StatSoft (Europe) GmbH, Hamburg, Germany) [42].

From the resulting data matrix, datasets were generated for each metal under study. Each dataset contains from 60 to 242 data tuples of the following types:Participant code—information field;Weight (kg)—positive real value;Height (cm)—positive real value;Body area (m^2^)—positive real value;Age (years)—positive integer value;Gender (0-female, 1-male)-boolean value;Metal in drinking water (mg/L)—real positive value;Metal in the soil (mg/kg)—material positive value;Metal retention (unit)—positive real value in the interval (0;1).

Each dataset was split into 80% and 20% training and test sets, respectively [36]. All data were normalized to the interval [−1;1]. We selected the number of layers of MLP neural networks, the number of neurons in each layer, and the type of activation function for each layer by conducting a series of exploratory experiments with variations of the specified parameters in the ranges specified by experts. The learning rate remained constant and was equal to 0.5 for all experiments, corresponding to the default settings for the modeling neural network package and ensuring the required calculation accuracy.

### 3.1. Building an Open Double-Loop Neural Network Model

The two blocks of the model input the morpho-physiological parameters of people inhabiting a given location (height, weight, sex, and age) and metal concentrations in external environments that determine the long-term or short-term nature of metal exposure to the organism—drinking water and soil. The model’s output will be the retention of the corresponding metal in the organism. We conducted modeling based on the Loginom 7.2 analytical platform.

#### 3.1.1. Building a Neural Network Block for Calculating the Long-Term Effects of Metals on the Human Body

The block is an MLP-type feed-forward neural network [36] with the following characteristics (selected experimentally):Number of neurons of the input layer—7;Number of hidden layers—1;Number of neurons in the hidden layer—5;The hidden layer activation function is a hyperbolic tangent with curvature parameter 0.7;Number of neurons of the output layer—1;The activation function of the output neuron is linear.

The training algorithm for all models was resilient propagation (RPROP) with a descent step of 0.5 and a split factor of 1.2, respectively. Each model was trained on its own training dataset for 10,000 epochs.

A general view of the topology of the models is shown in Figure 4, using zinc as an example.

The technology allows us to calculate the level of retention in the body of four major metals: copper, zinc, lead, and chromium.

The input data are fed to the input of the pre-trained model. The output of the model is the retention value of the investigated metal for a particular person. The model can be pre-trained on the newly obtained data. Pre-training on the retention data of residents of a particular region will lead to the adaptation of the technology to the local conditions of the study area.

Training error ${Err}_{i}^{O}$ was calculated for each *i-th* training example as the squared deviation of the output and reference values of the output parameter (retention) normalized to 1:(4)$${Err}_{i}^{O}={\left(y_{i}^{M}-y_{i}^{e}\right)}^{2}$$
where $i=\bar{1,N}$, *N*—number of training examples, $y_{i}^{M}$—normalized output of the neural network model for the *i-th* example, $y_{i}^{e}$—normalized reference response for the *i-th* example.

As the overall training error of the model ${Err}_{M}^{O}$ calculation of the average normalized error over the whole data set was applied:(5)$${Err}_{M}^{O}=\frac{1}{N}\sum_{i=1}^{N}{Err}_{i}^{O}$$

Calculate neural network errors on the test set using similar formulas:(6)$${Err}_{j}^{T}={\left(y_{j}^{M}-y_{j}^{e}\right)}^{2}$$
(7)$${Err}_{M}^{T}=\frac{1}{K}\sum_{j=1}^{K}{Err}_{j}^{T}$$
where $j=\bar{1,K}$, *K*—number of test examples.

#### 3.1.2. Building a Neural Network Block for Calculating the Short-Term Effects of Metals on the Human Body

Just as for the long-term exposure block, the short-term exposure block is an MLP-type neural network. The network characteristics are as follows:Number of neurons of the input layer—6;Number of hidden layers—1;Number of neurons in the hidden layer—5;The hidden layer activation function is a hyperbolic tangent with curvature parameter 0.7;Number of neurons of the output layer—1;The activation function of the output neuron is linear.

The training algorithm for all models was resilientpropagation (RPROP) with a descent step of 0.5 and a split factor of 1.2, respectively. Each model was trained on its own training dataset for 10,000 epochs.

A general view of the topology of the models is shown in Figure 5 (using the zinc retention model as an example).

Training error ${Err}_{i}^{O}$ was calculated for each *i-th* training example by formula (4). The total block training error for each metal ${Err}_{M}^{O}$ was calculated by Formula (5). Calculate neural network errors ${Err}_{j}^{T}$ and ${Err}_{M}^{T}$ on the test set using the Formulas (6) and (7).

Both neural network models were trained separately on their own data sets. The constructed open-ended two-loop neural network model for the estimation of metal retention in the organism has the following form, see Figure 6:

### 3.2. Building a Closed Double-Loop Neural Network Model

In contrast to the open double-loop model of metal exposure assessment, the closed-loop model has a single output for the long-term and short-term exposure blocks. The output is calculated using an additional mathematical operations to calculate the weighted average of the outputs of both blocks.

It is proposed that the average error of each neural network block be used on the test data to calculate the weights. The weights, in this case, can be considered as generalized confidence coefficients in the predictions of each of the neural network blocks:

$a_{M}^{l}$—confidence coefficient of the long-term impact unit;

$a_{M}^{s}$× confidence coefficient of the short-term exposure unit.
(8)$$a_{M}^{l}=\frac{{Err}_{M}^{T}(l)}{{Err}_{M}^{T}\left(l\right)+{Err}_{M}^{T}\left(s\right)}$$
(9)$$a_{M}^{s}=\frac{{Err}_{M}^{T}(s)}{{Err}_{M}^{T}\left(l\right)+{Err}_{M}^{T}\left(s\right)}$$

Here:

${Err}_{M}^{T}(l)$—average testing error of the neural network block of long-term exposure for metal *M*.

${Err}_{M}^{T}(s)$—average testing error of the neural network block of short-term exposure for metal *M*.

Then, for each metal, the output of the closed-loop two-loop model will be calculated according to the following formula:(10)$${Retention}_{M}={a_{M}^{l}Retention}_{M}^{l}+{a_{M}^{s}Retention}_{M}^{s}$$

Here:

${Retention}_{M}$—an overall assessment of *M* metal retention, taking into account both long-term and short-term impacts;

${Retention}_{M}^{l}$—retention estimation according to the neural network block of long-term effects for metal *M*;

${Retention}_{M}^{s}$—retention estimation according to the neural network block of short-term exposure for metal *M*.

The structure of the two-loop closed-loop neural network model of retention estimation in the organism, taking into account the period of exposure, will be as follows, see Figure 7:

Thus, the closed-loop model can be finally built only after the individual blocks of the open-loop model have been built and tested.

## 4. Experiments and Results

To verify the effectiveness of the developed model, we conducted experiments on test data for long-term and short-term exposure neural network blocks.

### 4.1. Results of Computational Experiments for the Open Double-Loop Model

Experiments on the open-loop model were conducted separately for the long-term and short-term impact blocks.

#### 4.1.1. Results of Computational Experiments for the Long-Term Impact Block

A graphical representation of the difference between the reference and calculated retention values is presented in Figure 8 (training examples are ordered in ascending order of the reference metal retention values).

The training results for the considered metals are presented in Table 1.

A graphical representation of the root mean square (RMS) errors of the neural network blocks on the test datasets in Figure 9.

The accuracy of the test data is presented in Table 2.

#### 4.1.2. Results of Computational Experiments for the Short-Term Impact Block

A graphical representation of the difference between the reference and calculated retention values is presented in Figure 10 (training examples are ordered in ascending order of the reference metal retention values).

The training results for the considered metals are presented in Table 3.

A graphical representation of the RMS errors on the test datasets is presented in Figure 11.

The accuracy of the test data is presented in Table 4.

### 4.2. Results of Computational Experiments for the Closed Double-Loop Model

Based on the experiments described in Section 3.1.1 and Section 3.1.2, confidence coefficients were determined for the practical implementation of the closed-loop two-loop neural network model. The results of calculating the confidence coefficients for the considered metals are summarized in Table 5.

For the closed double-loop system, we conducted experiments on the full data set. Figure 12 and Figure 13 present the graphical results of the calculations. The error values for the closed model are shown in Table 6.

The benefit of using a closed double-loop model compared to an open-loop model is clearly shown in Figure 14.

We achieved an increase in accuracy in almost all cases. The long-term model for calculating copper retention was the exception—its accuracy almost coincides with the closed double-loop model. We achieved the maximum improvement for calculating chromium retention.

To verify the initial conclusions about the efficiency and adequacy of the designed models, we used a 5-fold cross-validation test. We created five copies for each neural network model, which we designed and trained initially. The synaptic weights of the copy models were reset to small random values, after which each copy model was trained on the original data set, divided in the proportion of 80% and 20% into the training and test sets, respectively, with the elements of the test set selected by a sequential shift of 20% of the total sample, starting with the first element. The results were assessed by the number of correctly predicted retention levels on the test set. The results showed a consistently high accuracy of all designed models.

## 5. Discussion and Conclusions

The results of the experiments with the block of long-term impacts on the training set demonstrate the potential of our models. The maximum error is obtained for the calculated block of lead. The results of testing the models show that the accuracy of calculations of test data for all investigated metals is at least 90%. For copper, the calculation error for the test examples turned out to be the smallest. Together with the modeling results on the training set, the neural network block for copper calculation has the highest degree of adequacy. The test errors of the blocks for chromium and lead are approximately the same (about 10%). However, the training error of chromium calculation is an order of magnitude lower than that of lead. This circumstance may indicate the need for retraining the chromium calculation block, highlighting a potential limitation of the models that warrants further investigation.

The results of experiments with the short-term exposure block on the training set demonstrate that the standard topology of the models provides a learning accuracy of at least 97% for the whole set of metals under study. Comparative results of training errors generally correspond to those for the long-term block: lead introduces the maximum error. The results of experiments on test data show that the accuracy of calculations for all metals under study does not fall below 92%.

After testing two blocks of the open double-loop model, we calculated the confidence coefficients in the forecast of these models and built a closed double-loop model. Experiments showed that this model’s accuracy is even higher than that of individual long-term and short-term impact models. The benefit in accuracy ranged from 7 times compared with the short-term model to 10 times compared with the long-term model.

Our proposed method of assessment and quantification of metal retention in the human body represents a significant advancement in our field. It offers a high degree of accuracy and reliability without the need for expensive laboratory tests. This method allows us to quantify the level of retention in the body based on readily available information, making it a practical and efficient tool for our colleagues in environmental and human health research. It is important to note that the accumulation of metals in the body of children and teenagers is not only a complex indicator covering all pathways of metal intake into the body but also a long-term prognostic criterion with significant inertia and aggregating information over a long period of time. The results of this study are intended for patient-oriented diagnostics and treatment of environmentally induced microelements in the territory of individual megacities. The developed model can be used in various fields of ecology and health care to predict the condition of individuals and entire communities, assess and forecast the ecological well-being of the territory from the point of view of health preservation, and obtain a comprehensive forecast both in the short and long term. Figure 15 shows comparative characteristics of the accuracy of the open and closed models. 

As with any technology for constructing a data-based model, our approach is sensitive to data, and the accuracy and efficiency of the model depend on its representativeness. The consequence is the limited application scope—each configured model will be effective only for the territory where the data were obtained. A possible solution to this problem may be expanding the training data set due to other population groups and territories. With this approach, to ensure the adequacy of the model for the entire generalized set, the use of hybrid neural network models that perform preliminary clustering of input data with subsequent final processing by an ensemble of two-loop neural network models will be promising.

Another interesting direction for future research is to increase the interpretability of the model response, bringing the response closer to natural human language. The use of neuro-fuzzy networks and fuzzy logical inference models for these purposes may have great potential, along with modern visualization tools. For the widespread implementation of the proposed neural network approach based on the two-loop model, it is advisable to develop unique software in decision support systems. Specialists should solve this task by considering the market’s needs based on the ergonomic requirements of the customer, and it can also be a direction for further research and development in this area.

## Figures and Tables

**Figure 1 sensors-24-07157-f001:**
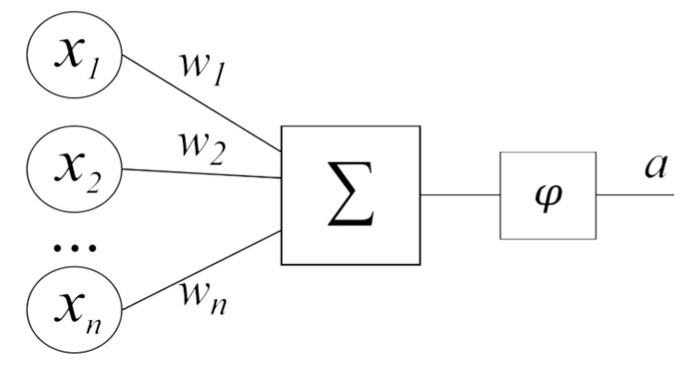
Artificial neuron model.

**Figure 2 sensors-24-07157-f002:**
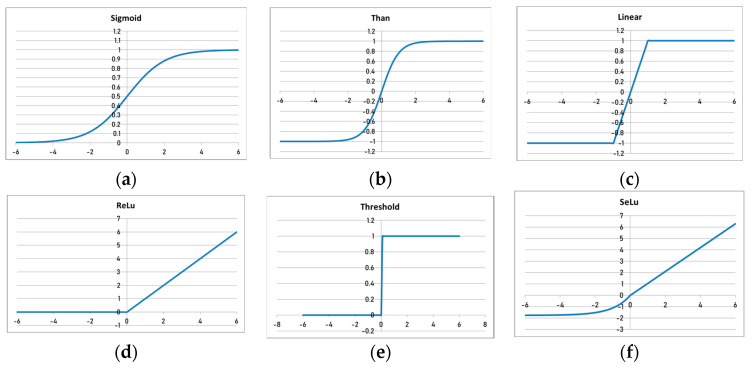
Main types of activation function of artificial neuron: (**a**) Sigmoid, (**b**) Hyperbolic tangent, (**c**) Linear, (**d**) ReLU, (**e**) Threshold, (**f**) SeLU.

**Figure 3 sensors-24-07157-f003:**
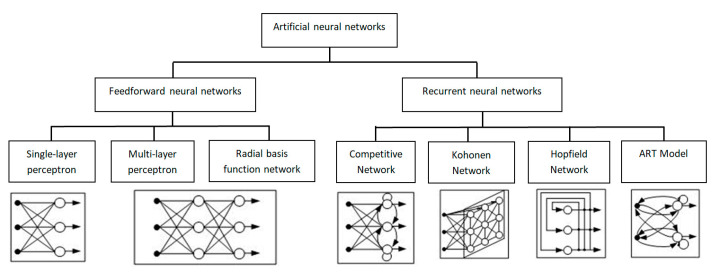
Types of neural networks.

**Figure 4 sensors-24-07157-f004:**
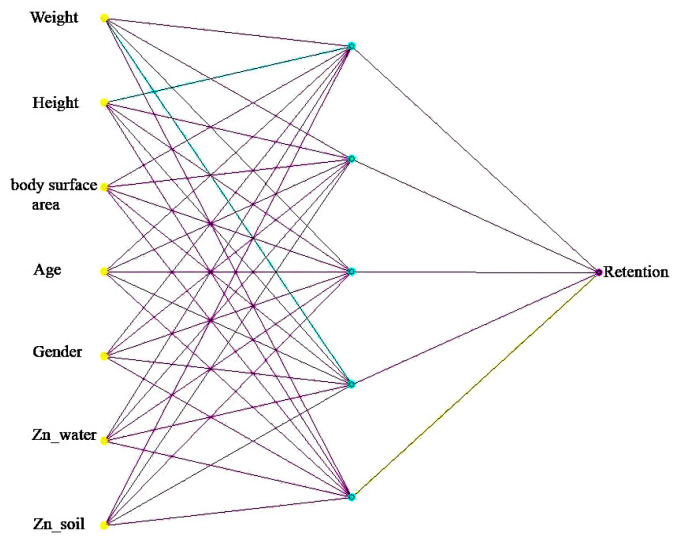
Neural network architecture of a block for calculating long-term impact on zinc concentrations.

**Figure 5 sensors-24-07157-f005:**
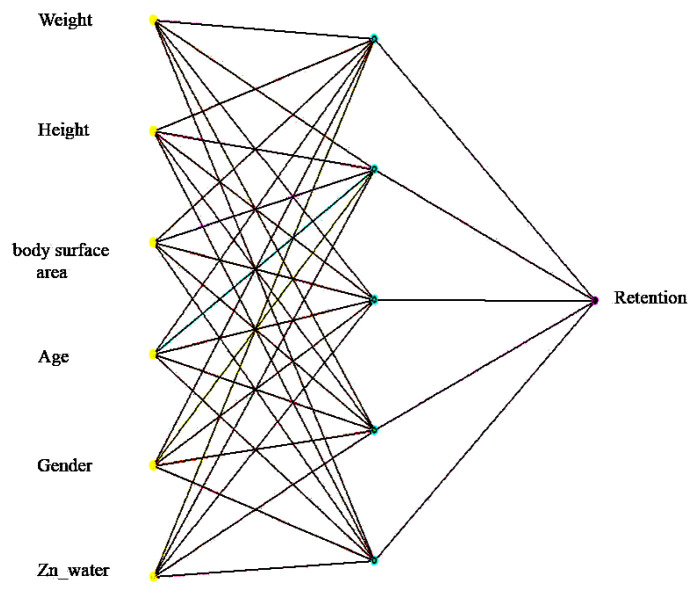
Neural network architecture of the block for calculating short-term impact on zinc concentrations.

**Figure 6 sensors-24-07157-f006:**
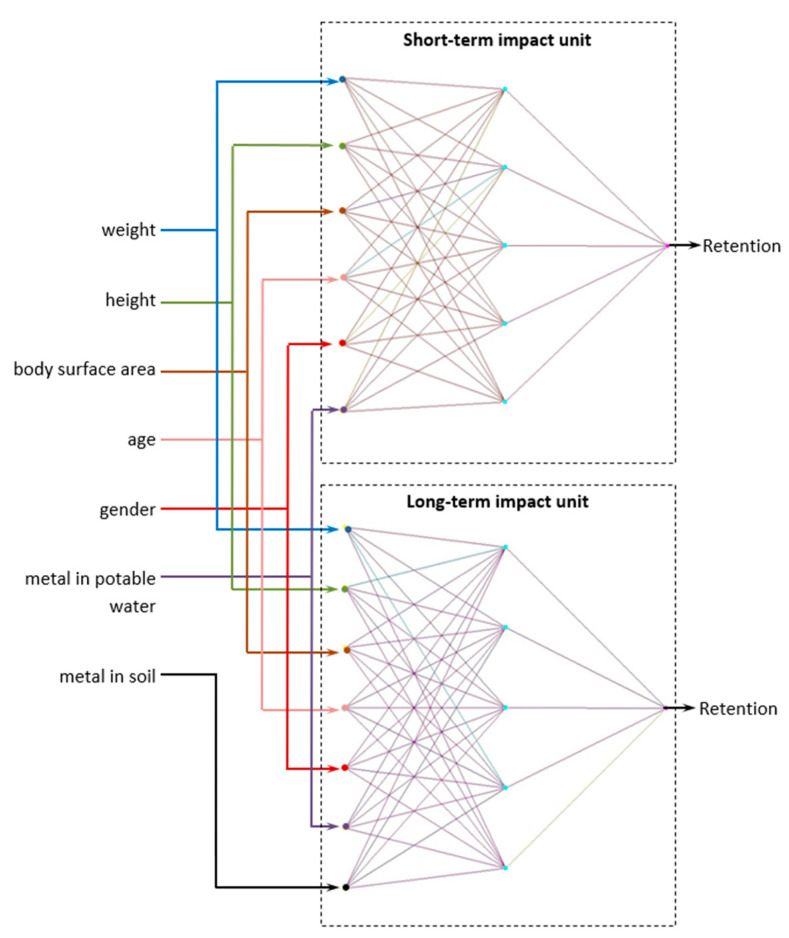
Structural diagram of a two-loop open-loop neural network model for estimating metal retention in the body taking into account the period of exposure.

**Figure 7 sensors-24-07157-f007:**
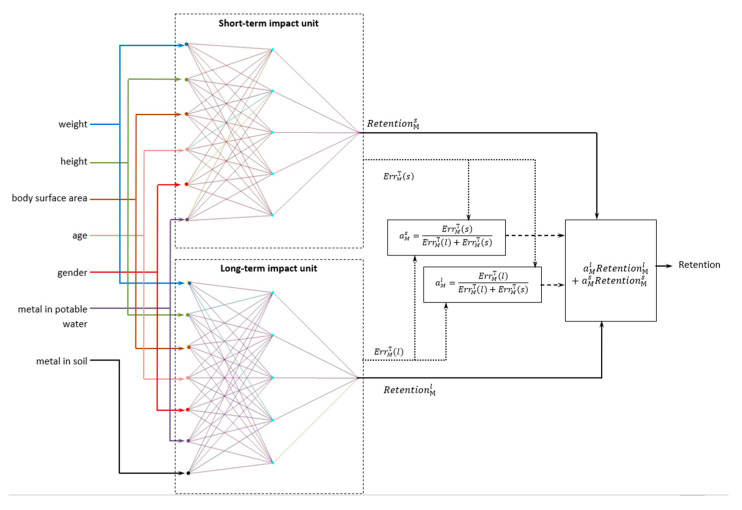
Structural diagram of a two-loop closed-loop neural network model for estimating metal retention in the body, taking into account the period of exposure based on the calculation of confidence coefficients.

**Figure 8 sensors-24-07157-f008:**
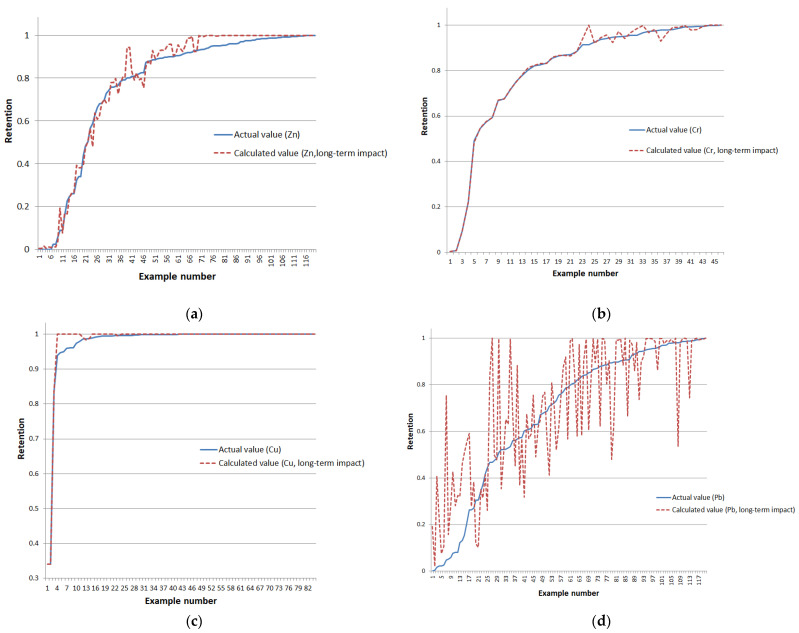
Deviation of reference and model-calculated retention values for long-term impact type on training examples of (**a**) zinc, (**b**) chromium, (**c**) copper, and (**d**) lead.

**Figure 9 sensors-24-07157-f009:**
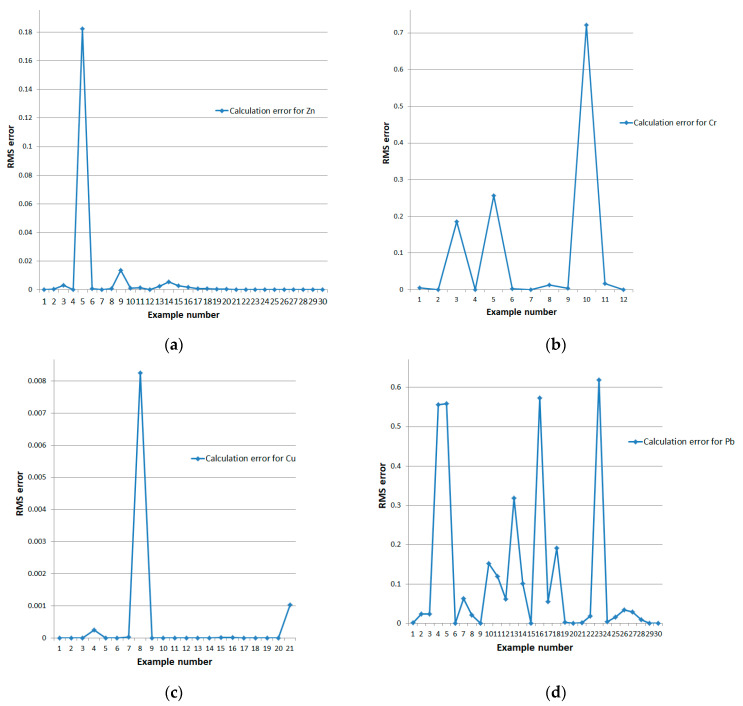
RMS errors of neural network models of standard topology on test datasets for long-term impact type: (**a**) zinc, (**b**) chromium, (**c**) copper, and (**d**) lead.

**Figure 10 sensors-24-07157-f010:**
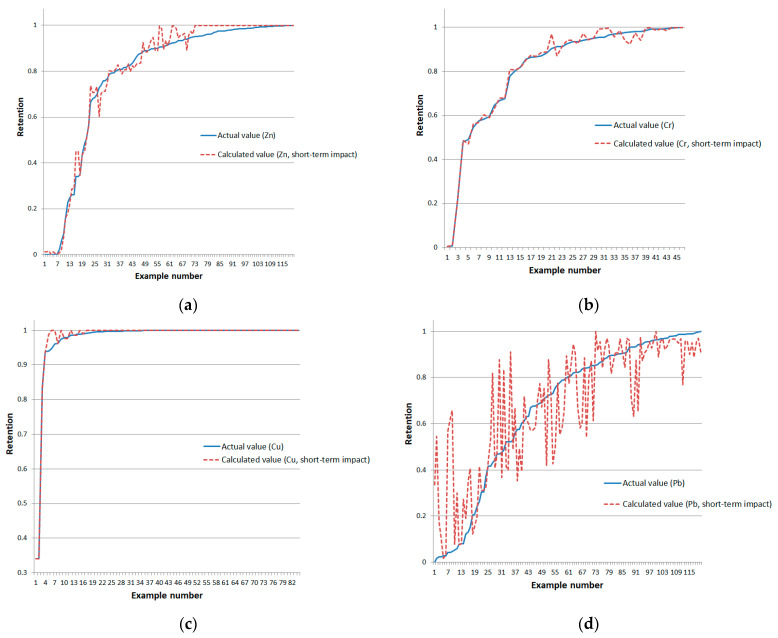
Deviation of reference and model-calculated retention values for short-term impact type on training examples of (**a**) zinc, (**b**) chromium, (**c**) copper, and (**d**) lead.

**Figure 11 sensors-24-07157-f011:**
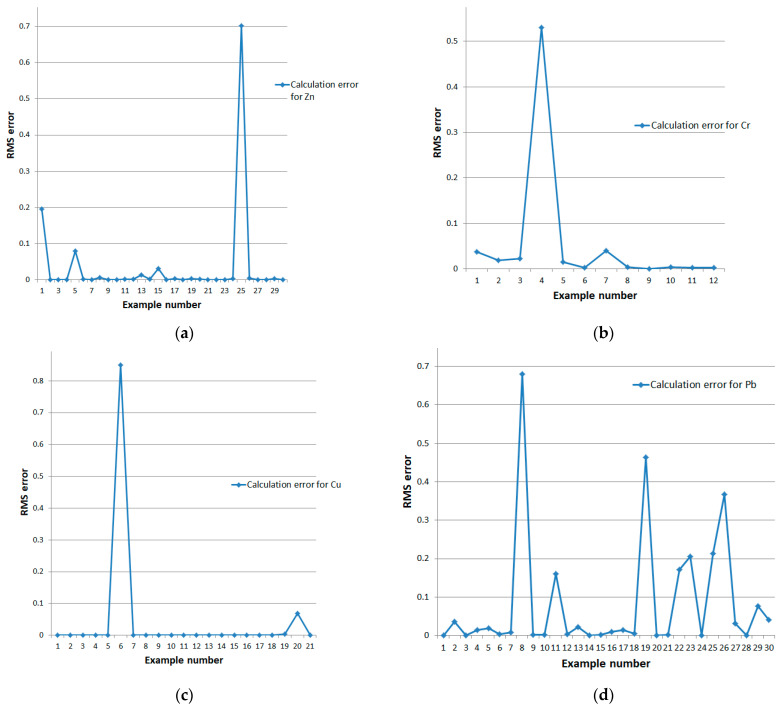
RMS errors of neural network models of typical topology on test datasets for short-term exposure type on examples of (**a**) zinc, (**b**) chromium, (**c**) copper, and (**d**) lead.

**Figure 12 sensors-24-07157-f012:**
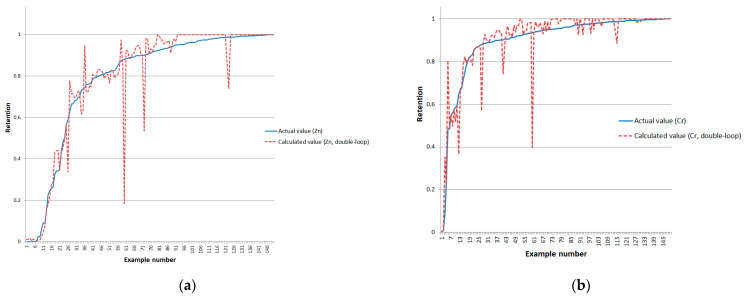
Deviation of reference and model-calculated retention values for a closed double-loop system for the full dataset of (**a**) zinc, (**b**) chromium, (**c**) copper, and (**d**) lead.

**Figure 13 sensors-24-07157-f013:**
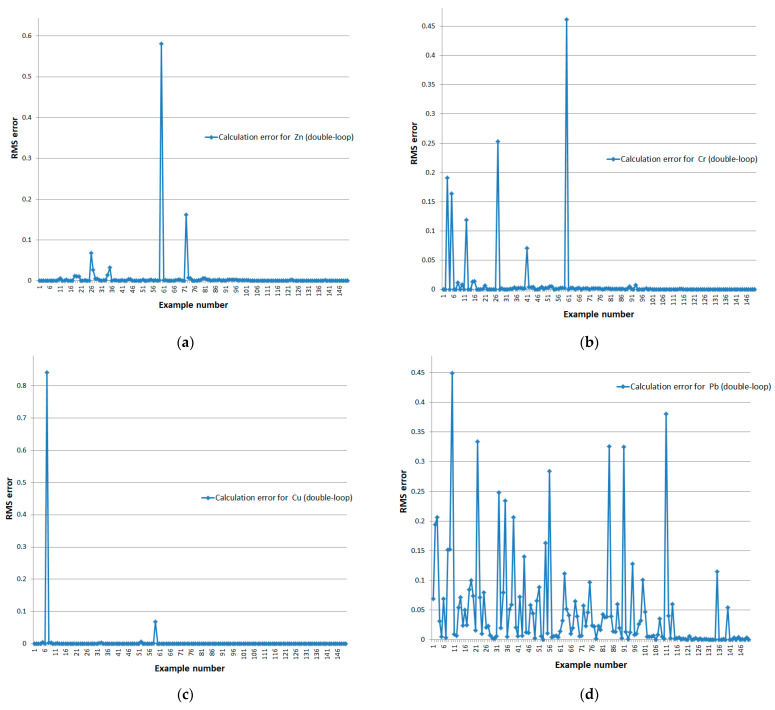
RMS errors of a closed double-loop system for the full dataset: (**a**) zinc, (**b**) chromium, (**c**) copper, and (**d**) lead.

**Figure 14 sensors-24-07157-f014:**
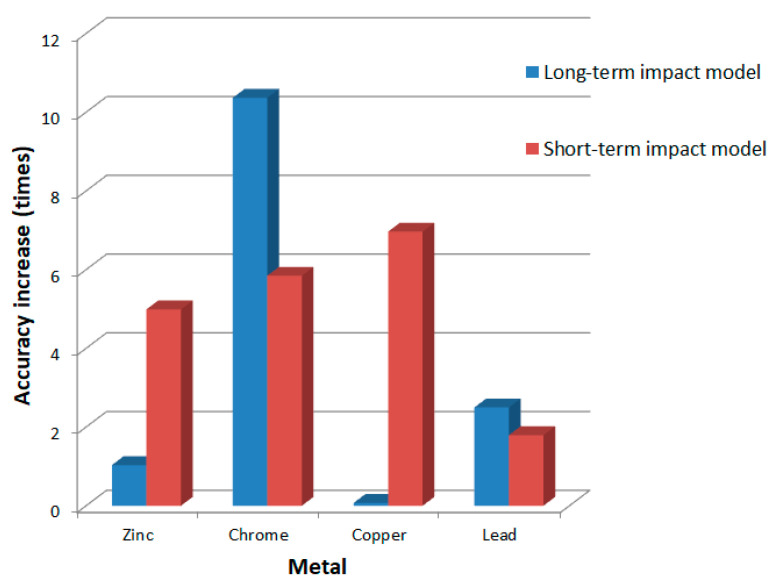
The relative increase in the accuracy of retention calculations when using a double-loop model relative to long-term and short-term impact models.

**Figure 15 sensors-24-07157-f015:**
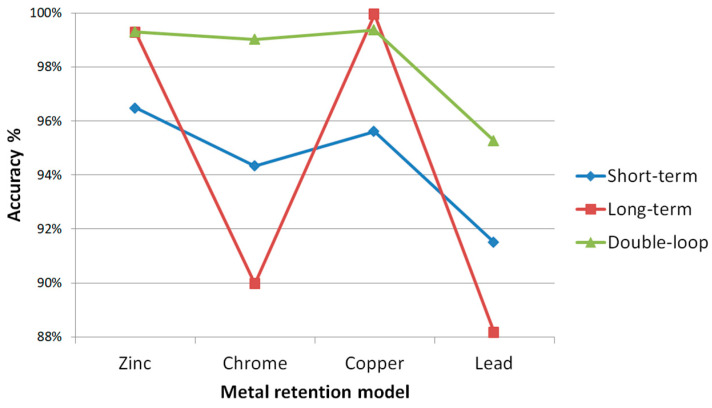
Comparative accuracy of the developed models.

**Table 1 sensors-24-07157-t001:** Accuracy characteristics of the designed neural network models of typical topology on training datasets for long-term type of exposure.

Metal Retention Model	Minimum Error on Training Examples ${min(Err}_{i}^{O})$	Maximum Error on Training Examples ${max(Err}_{i}^{O})$	Total Model Training Error ${Err}_{M}^{O}$
Zinc	0.0	0.0209	0.0015
Chrome	0.0	0.0074	0.0003
Copper	0.0	0.0086	0.0004
Lead	0.0	0.5034	0.0355

**Table 2 sensors-24-07157-t002:** Accuracy characteristics of the designed neural network models of typical topology on test datasets for the long-term type of impact.

Metal Retention Model	Minimum Error on Test Cases ${min(Err}_{j}^{T})$	Maximum Error by Test Examples ${max(Err}_{j}^{T})$	Total Model Testing Error ${Err}_{M}^{T}$
Zinc	3.5142 × 10^−6^	0.1823	0.0073
Chrome	6.5846 × 10^−6^	0.7216	0.1003
Copper	1.9207 × 10^−11^	0.0082	0.0005
Lead	1.1487 × 10^−7^	0.6184	0.1183

**Table 3 sensors-24-07157-t003:** Accuracy characteristics of the designed neural network models of typical topology on training datasets for short-term impact type.

Metal Retention Model	Minimum Error on Training Examples ${min(Err}_{i}^{O})$	Maximum Error on Training Examples ${max(Err}_{i}^{O})$	Total Model Training Error ${Err}_{M}^{O}$
Zinc	2.2816 × 10^−28^	0.0161	0.0011
Chrome	5.6740 × 10^−17^	0.0043	0.0005
Copper	0	0.0065	0.0002
Lead	1.4691 × 10^−9^	0.3729	0.0297

**Table 4 sensors-24-07157-t004:** Accuracy characteristics of the designed neural network models of typical topology on test datasets for short-term impact type.

Metal Retention Model	Minimum Error on Test Examples ${min(Err}_{j}^{T})$	Maximum Error on Test Examples ${max(Err}_{j}^{T})$	Total Model Testing Error ${Err}_{M}^{T}$
Zinc	2.0306 × 10^−5^	0.7015	0.0351
Chrome	2.4245 × 10^−5^	0.5301	0.0565
Copper	3.4485 × 10^−10^	0.8501	0.0439
Lead	5.4239 × 10^−7^	0.6803	0.0849

**Table 5 sensors-24-07157-t005:** Confidence weights for the blocks of long-term and short-term body metal exposure.

Metal	${Err}_{M}^{T}(l)$	${Err}_{M}^{T}(s)$	$a_{M}^{l}$	$a_{M}^{s}$
Zinc	0.0073	0.0351	0.1718	0.8281
Chrome	0.1003	0.0565	0.6394	0.3605
Copper	0.0004	0.0439	0.0102	0.9897
Lead	0.1183	0.0849	0.5821	0.4178

**Table 6 sensors-24-07157-t006:** Accuracy characteristics of the designed closed double-loop system for the full dataset.

Metal Retention Model	Minimum Error	Maximum Error	Total Model Error
Zinc	5.6 × 10^−19^	0.5809	0.0070
Chrome	2.08 × 10^−18^	0.4614	0.0096
Copper	0	0.8414	0.0063
Lead	2.27 × 10^−7^	0.4488	0.0473

## Data Availability

The data presented in this study are available on request from the corresponding author. The data are not publicly available due to the rules of our contract conditions with our customers.

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
