# Peer review of "Advanced Low-Cost Technology for Assessing Metal Accumulation in the Body of a Metropolitan Resident Based on a Neural Network Model"

_sensors, 2024, doi:10.3390/s24227157_

Round 1

Reviewer 1 Report

Comments and Suggestions for Authors

1. A relatively long abstract that contains uninteresting sentences, it can be improved.

2. Could you please improve the quality of Figures 1 and 2?

3. Please explain further the characteristics obtained in Figure 9 (d).

4. Can you discuss the limitations of your proposed methods in future work (at the end of the conclusion)?

Comments on the Quality of English Language

- The writing is good, however some grammatical errors that need to be addressed.

Author Response

We sincerely thank the reviewer for his attentiveness to our article. We tried to consider and correct all the comments, which certainly allowed us to significantly improve our work.

  1. A relatively long abstract that contains uninteresting sentences, it can be improved.

Thank you for your comment. We have revised and shortened the abstract.

  1. Could you please improve the quality of Figures 1 and 2?

The quality of the figures has been improved by increasing brightness and contrast.

  1. Please explain further the characteristics obtained in Figure 9 (d).

The results of the lead retention modeling are shown in Figure 9 (d). Unlike other metals, lead shows unstable retention rates. This may indicate that the neural network model for lead should be revised in the future, supplemented with preliminary filters. We also emphasized this fact in Section 5, Discussion and Conclusions.

  1. Can you discuss the limitations of your proposed methods in future work (at the end of the conclusion)?

At the end of Section 5, Discussion and Conclusions (previously Section 4), we have added a description of our models' limitations and outlined ways to further improve the proposed methodology and possible areas of its application.

Reviewer 2 Report

Comments and Suggestions for Authors

This paper presents a novel approach to assessing metal accumulation in the bodies of urban residents using neural network models. It is a valuable contribution to the field and has the potential to address a significant public health concern. The paper is well-structured, logically organized, and presents a clear and compelling argument. However, there are some areas where further development is needed.

1. Add a dedicated section on the theoretical foundation of neural network models. Explain the basic principles, including different architectures (e.g., MLP, CNN, RNN), activation functions (e.g., sigmoid, ReLU, tanh), and training algorithms (e.g., backpropagation, gradient descent).

2. Justify the choice of specific model architectures (e.g., open-loop vs. closed-loop) based on the characteristics of the problem. Discuss how these choices align with the goals of the study, such as capturing short-term and long-term effects, and improving prediction accuracy.

3. Provide a rationale for the specific parameter settings used in the models, such as the number of neurons, hidden layers, and learning rate. Discuss how these parameters were determined and their potential impact on model performance.

4. Mention efforts made to ensure the representativeness and diversity of the study population, such as stratified sampling or inclusion of participants from different neighborhoods. Discuss potential limitations due to the single-source data and propose strategies for addressing them, such as incorporating data from other regions.

5. Provide more details on the preprocessing steps, including data cleaning, normalization, and feature selection. Discuss any assumptions made and their justification. Explain the rationale behind the chosen features and their potential impact on model performance.

6. Present the results in a structured and visually appealing manner using tables, figures, and charts. Clearly summarize key findings, such as prediction accuracy, error metrics, and model comparisons.  Compare the performance of the models across multiple metrics, including predictive accuracy, interpretability, computational efficiency, and ease of implementation. Discuss the trade-offs between these metrics and the implications for real-world applications.

7. Address the limitations of the study, such as the single-source data and narrow application scope. Discuss potential strategies for overcoming these limitations, such as incorporating data from other regions and population groups. Model Interpretability: Enhance the interpretability of the models by discussing potential mechanisms and factors influencing metal accumulation. Consider using techniques like feature importance analysis, visualization, or sensitivity analysis to gain insights into the model's predictions. Application Scope: Broaden the discussion on the application scope by exploring potential uses of the models in clinical diagnostics, environmental monitoring, and public health interventions. Discuss the implications of the findings for these applications and suggest future research directions.

8. Summarize the main contributions of the study, emphasizing the significance of the findings for the field of environmental health and the potential to improve the assessment and management of metal accumulation. Highlight the limitations of the study and suggest potential avenues for future research, such as evaluating model generalizability, analyzing influencing factors, and exploring practical applications.

Additional Suggestions:

  • Implement cross-validation or leave-one-out validation to assess the model's performance on different data sets and ensure the robustness of the findings.
  • Conduct a thorough analysis of the factors influencing metal accumulation, including dietary habits, lifestyle, and environmental exposure. This could provide valuable insights into the underlying mechanisms and help identify potential interventions.
  • Develop software tools or mobile applications based on the models to facilitate their use by healthcare professionals and the public. This could enhance the practical applicability of the study and contribute to the dissemination of the findings.
Comments on the Quality of English Language

The quality of the English language in the given context appears to be satisfactory. The sentences are structured clearly and logically, making it easy for readers to understand the intended meaning. The vocabulary used is appropriate and contributes to the clarity of the text. There are no obvious grammatical errors or awkward phrasing that would detract from the reader's comprehension.

Author Response

We sincerely thank the reviewer for his attentiveness to our article. We tried to consider and correct all the comments, which certainly allowed us to significantly improve our work.

This paper presents a novel approach to assessing metal accumulation in the bodies of urban residents using neural network models. It is a valuable contribution to the field and has the potential to address a significant public health concern. The paper is well-structured, logically organized, and presents a clear and compelling argument. However, there are some areas where further development is needed.

  1. Add a dedicated section on the theoretical foundation of neural network models. Explain the basic principles, including different architectures (e.g., MLP, CNN, RNN), activation functions (e.g., sigmoid, ReLU, tanh), and training algorithms (e.g., backpropagation, gradient descent).

Section 2. Preliminary information has been added to the article, where we described the main theoretical points regarding the use of neural networks for forecasting tasks.

  1. Justify the choice of specific model architectures (e.g., open-loop vs. closed-loop) based on the characteristics of the problem. Discuss how these choices align with the goals of the study, such as capturing short-term and long-term effects, and improving prediction accuracy.

At the end of the new section 2. Preliminary information, we provided the rationale for choosing a multilayer perceptron as the main working model for our study.

  1. Provide a rationale for the specific parameter settings used in the models, such as the number of neurons, hidden layers, and learning rate. Discuss how these parameters were determined and their potential impact on model performance.

In Section 3. Materials and Methods (formerly Section 2), a paragraph has been added to the introductory part describing the procedure for determining the parameter settings based on a series of exploratory experiments.

  1. Mention efforts made to ensure the representativeness and diversity of the study population, such as stratified sampling or inclusion of participants from different neighborhoods. Discuss potential limitations due to the single-source data and propose strategies for addressing them, such as incorporating data from other regions.

In Section 5. Discussion and Conclusions (formerly Section 4), we added a paragraph explaining the binding of neural network models to a specific location. We also showed possible ways to overcome this drawback by creating hybrid models.

  1. Provide more details on the preprocessing steps, including data cleaning, normalization, and feature selection. Discuss any assumptions made and their justification. Explain the rationale behind the chosen features and their potential impact on model performance.

In Section 3. Materials and Methods (previously Section 2), we have added a description of the data pre-cleaning and processing steps and provided references to relevant further literature.

  1. Present the results in a structured and visually appealing manner using tables, figures, and charts. Clearly summarize key findings, such as prediction accuracy, error metrics, and model comparisons.  Compare the performance of the models across multiple metrics, including predictive accuracy, interpretability, computational efficiency, and ease of implementation. Discuss the trade-offs between these metrics and the implications for real-world applications.

We have added explanatory graphs to Section 5. Discussion and Conclusions (former Section 4). Further comparative study of accuracy by different criteria is the topic of our future research.

  1. Address the limitations of the study, such as the single-source data and narrow application scope. Discuss potential strategies for overcoming these limitations, such as incorporating data from other regions and population groups. Model Interpretability: Enhance the interpretability of the models by discussing potential mechanisms and factors influencing metal accumulation. Consider using techniques like feature importance analysis, visualization, or sensitivity analysis to gain insights into the model's predictions. Application Scope: Broaden the discussion on the application scope by exploring potential uses of the models in clinical diagnostics, environmental monitoring, and public health interventions. Discuss the implications of the findings for these applications and suggest future research directions.

At the end of Section 5. Discussion and Conclusions (previously Section 4), we have added a description of the limitations of our models, and outlined ways to further improve the proposed methodology and possible areas of its application.

  1. Summarize the main contributions of the study, emphasizing the significance of the findings for the field of environmental health and the potential to improve the assessment and management of metal accumulation. Highlight the limitations of the study and suggest potential avenues for future research, such as evaluating model generalizability, analyzing influencing factors, and exploring practical applications.

In Section 5. Discussion and Conclusions (previously Section 4), we have added a description of the limitations of our models, and outlined ways to further improve the proposed methodology and possible areas of its application.

Additional Suggestions:

  •  Implement cross-validation or leave-one-out validation to assess the model's performance on different data sets and ensure the robustness of the findings.

We tested the accuracy of our models using 5-fold cross-validation. A brief description of the experiments and their results is included in Section 4. Experiments and Results.

  • Conduct a thorough analysis of the factors influencing metal accumulation, including dietary habits, lifestyle, and environmental exposure. This could provide valuable insights into the underlying mechanisms and help identify potential interventions.

We have included a brief overview of the influencing factors in Section 1. Introduction. A more in-depth analysis and the resulting results will be used in further research.

  • Develop software tools or mobile applications based on the models to facilitate their use by healthcare professionals and the public. This could enhance the practical applicability of the study and contribute to the dissemination of the findings.

The development of software applications for practical use, including those based on mobile applications, is the topic of our further future research. We have included this information in Section 5. Discussion and Conclusions.

Reviewer 3 Report

Comments and Suggestions for Authors

The article is devoted to a very relevant topic - the assessment of the accumulation of metals in the body of a resident of a megalopolis. The authors propose a solution to this urgent and difficult environmental problem, the proposed neural network is significant advancement in this field.

The article is written clearly and in detail.

There is not any criticism except  the too high precision of accuracy characteristics (Tables 1 - 6).

Is there meaning of so many significant digits after the decimal point?

Author Response

We sincerely thank the reviewer for his attentiveness to our article. We tried to consider and correct all the comments, which certainly allowed us to significantly improve our work.

The article is devoted to a very relevant topic - the assessment of the accumulation of metals in the body of a resident of a megalopolis. The authors propose a solution to this urgent and difficult environmental problem, the proposed neural network is significant advancement in this field.

The article is written clearly and in detail.

There is not any criticism except  the too high precision of accuracy characteristics (Tables 1 - 6).

Is there meaning of so many significant digits after the decimal point?

Thank you for your comment. We have reduced the digit capacity of the tables to 4 decimal places.